# AXL Receptor in Cancer Metastasis and Drug Resistance: When Normal Functions Go Askew

**DOI:** 10.3390/cancers13194864

**Published:** 2021-09-28

**Authors:** Almira Auyez, A. Emre Sayan, Marina Kriajevska, Eugene Tulchinsky

**Affiliations:** 1Department of Biomedical Sciences, Nazarbayev University School of Medicine, Nur-Sultan 020000, Kazakhstan; almira.auyez@nu.edu.kz (A.A.); marina.kriajevska@nu.edu.kz (M.K.); 2Cancer Sciences Division, University of Southampton, Southampton SO16 6YD, UK; A.E.Sayan@soton.ac.uk; 3Department of Genetics and Genome Biology, University of Leicester, Leicester LE1 7RH, UK

**Keywords:** AXL, TAM receptors, epithelial-mesenchymal plasticity, drug resistance, metastasis

## Abstract

**Simple Summary:**

AXL is a member of the TAM (TYRO3, AXL, MER) family of receptor tyrosine kinases. In normal physiological conditions, AXL is involved in removing dead cells and their remains, and limiting the duration of immune responses. Both functions are utilized by cancers in the course of tumour progression. Cancer cells use the AXL pathway to detect toxic environments and to activate molecular mechanisms, thereby ensuring their survival or escape from the toxic zone. AXL is instrumental in controlling genetic programs of epithelial-mesenchymal and mesenchymal-epithelial transitions, enabling cancer cells to metastasize. Additionally, AXL signaling suppresses immune responses in tumour microenvironment and thereby helps cancer cells to evade immune surveillance. The broad role of AXL in tumour biology is the reason why its inhibition sensitizes tumours to a broad spectrum of anti-cancer drugs. In this review, we outline molecular mechanisms underlying AXL function in normal tissues, and discuss how these mechanisms are adopted by cancers to become metastatic and drug-resistant.

**Abstract:**

The TAM proteins TYRO3, AXL, and MER are receptor tyrosine kinases implicated in the clearance of apoptotic debris and negative regulation of innate immune responses. AXL contributes to immunosuppression by terminating the Toll-like receptor signaling in dendritic cells, and suppressing natural killer cell activity. In recent years, AXL has been intensively studied in the context of cancer. Both molecules, the receptor, and its ligand GAS6, are commonly expressed in cancer cells, as well as stromal and infiltrating immune cells. In cancer cells, the activation of AXL signaling stimulates cell survival and increases migratory and invasive potential. In cells of the tumour microenvironment, AXL pathway potentiates immune evasion. AXL has been broadly implicated in the epithelial-mesenchymal plasticity of cancer cells, a key factor in drug resistance and metastasis. Several antibody-based and small molecule AXL inhibitors have been developed and used in preclinical studies. AXL inhibition in various mouse cancer models reduced metastatic spread and improved the survival of the animals. AXL inhibitors are currently being tested in several clinical trials as monotherapy or in combination with other drugs. Here, we give a brief overview of AXL structure and regulation and discuss the normal physiological functions of TAM receptors, focusing on AXL. We present a theory of how epithelial cancers exploit AXL signaling to resist cytotoxic insults, in order to disseminate and relapse.

## 1. Introduction

Metastatic propensity and drug resistance are two fundamental features of cancer, which cause treatment failure and death of cancer patients. These two key features are intrinsically related. A proportion of cancer cells (so-called drug-tolerant persisters or DTPs) survive exposure to a drug and give rise to cancer relapse. On the other hand, during the metastatic process, cells are likely to utilize the same basic survival mechanisms to adapt to the hostile environment of foreign tissues. Survival and invasion are dependent on signaling pathways acting downstream of various surface receptors, including receptor tyrosine kinases (RTKs). In many cases, gain-of-function mutations in RTK-encoding genes, and their amplification or overexpression are associated with metastatic propensity and drug resistance. In various cancer types, patients’ survival is improved by drugs targeting RTKs as mono- or combination therapy. Understanding the biology of RTKs in normal and cancerous tissues is essential for tailoring therapeutic approaches. The TAM family of RTKs has emerged as an important driver of cancer, and among this family, AXL has attracted the most attention as a molecule when expressed possesses prognostic value in several cancer types, including breast, ovarian, lung, and pancreatic cancer [1,2]. Moreover, preclinical studies have convincingly demonstrated AXL involvement in metastasis and resistance to various anti-cancer agents (see below). Several ongoing clinical trials have investigated the therapeutic benefits of AXL inhibition [3,4]. The exclusive role of AXL in cancer progression is likely associated with AXL normal function in clearance of apoptotic cells and immunosuppression. In addition, the implication of AXL in epithelial cell plasticity in tumours explains its role in drug resistance and metastasis. Here, we discuss AXL function in normal homeostasis and review the current literature on the underlying mechanisms to explain the distinctive role of this receptor in cancer biology.

## 2. Overview of the Structure and Regulation of AXL Receptor

### 2.1. Structural Features of TAM Receptors and Their Interactions with the Ligands

AXL and two related proteins: TYRO3 and MER are single-pass transmembrane receptors. AXL ectodomain contains two fibronectin type III-like and two immunoglobulin-like repeats, with the latter responsible for the interactions with the ligands, Growth Arrest-Specific protein 6 (GAS6) and Protein S1 (PROS1) (Figure 1). As it will be discussed later, interactions between ligands and phosphatidylserine (PtdSer) are required for the full activation of TAM receptors and their functions in normal tissues and tumour microenvironment. In addition to GAS6 and PROS1, three other proteins have been identified as potential ligands of TAM receptors, Galectin-3, TUBBY, and TUBBY-like protein 1 (TULP1) [5,6,7]. However, their role in physiological processes regulated by TAM receptors is currently underexplored. The homology is not evenly distributed over TAM protein sequences; it is very high within the kinase domains (60–66% identity) and much lower in the ectodomains (30–35% identity). Structural differences in ectodomains of TAM receptors reflect their functional divergence and different affinities to the ligands. It has been broadly accepted that PROS1 binds TYRO3 and MER only, and GAS6 interacts with all three TAM proteins with the highest binding affinity to AXL [8]. However, a recent study has shown that PROS1 binds and activates AXL in glioma spheroids [9].

### 2.2. AXL Receptor Regulation by Proteolytic Cleavage

AXL expression is regulated at the level of gene transcription, by miRNA (will be discussed in the following sections), and at the posttranslational level by ectodomain shedding. The receptor is cleaved by proteolytic enzymes, ADAM10 and ADAM17, in the area immediately adjacent to the transmembrane helix, leading to the release of ~80 kDa ectodomain into the extracellular milieu [10,11]. The shed ectodomain, termed soluble AXL (sAXL), can bind AXL ligand GAS6 and provide negative feedback by interfering with AXL activation. In addition, AXL receptor is a substrate for γ-secretase. After shedding the extracellular part, γ-secretase cleaves the remaining part within the transmembrane domain, releasing intracellular AXL fragment. This fragment can be transported to the nucleus via a nuclear localization signal (NLS) located next to the transmembrane domain [10]. Nuclear AXL was detected in cultured non-small cell lung cancer cells [12], in schwannoma and melanoma samples by immunohistochemistry [13,14]. Recently, a direct interaction between AXL and *TP53* gene promoter DNA has been reported to repress *TP53* gene transcription in mesothelioma cells [15]. The mechanism of incorporation of nuclear AXL into chromatin surrounding *TP53* promoter remains obscure. Although NLS adjacent to the transmembrane domain is present in TYRO3, but not in MER. Nuclear localization of either receptor was reported in leiomyosarcoma and acute lymphoblastic leukemia, respectively [16,17].

## 3. Overview of AXL-Regulated Signaling 

### 3.1. Docking Sites for Signaling Proteins

Binding a ligand results in AXL receptor dimerization and cross-phosphorylation at several tyrosine residues positioned within the kinase autoregulatory loop (Y698, Y702 and Y703) and distal portion of the cytoplasmic domain (Y779, Y821 and Y866) (Figure 1) [18,19]. The impact of the autoregulatory loop phosphorylation has not been investigated in the context of AXL, but mutating homologous residues to phenylalanines, within MER, strongly repressed or entirely abrogated protein kinase activity [20]. Phosphorylation at Y779, Y821 and Y866 generated docking sites attracting signaling molecules with SH2 domains. Specifically, phosphorylation at Y821 recruited a number of effectors, including phospholipase Cγ, phosphatidylinositol 3-kinase (PI3K) regulatory subunits p85α and β, cellular tyrosine kinase c-SRC (SRC), lymphocyte-specific protein tyrosine kinase (LCK) and growth factor receptor-bound protein 2 (GRB2). Phosphorylated Y779 or Y866 represented docking sites for p85α/β and phospholipase C, respectively [19] (Figure 1). Therefore, the engagement of AXL receptors may lead to the activation of critical pathways implicated in cell survival and proliferation, mitogen- activated protein kinases (MAPK), PI3K, protein kinase C (PKC), and SRC.

### 3.2. AXL Dimerization Partners and Diversification of Downstream Signaling

In addition to homodimerization, AXL receptors are able to heterodimerize with TYRO3, leading to cross-phosphorylation and activation of downstream PI3K and MAPK/ERK signaling pathways [21,22]. AXL also interacts with the more distantly related receptors, such as epidermal growth factor receptor (EGFR), human epidermal growth factor receptor 2 (HER2), tyrosine-protein kinase Met (cMET), and platelet-derived growth factor receptor (PDGFR) [23,24,25,26,27]. The data indicate that AXL may trigger phosphorylation of other receptors and stimulate downstream signaling in a cell type-specific fashion. For example, in esophageal squamous cell carcinoma cells, AXL directly binds and phosphorylates EGFR in an EGF-independent mode. This leads to the activation of the mechanistic target of rapamycin (mTOR) via PKC without PI3K/AKT involvement. Subsequently, AXL mediates a bypass of the PI3K-AKT module, leading to the development of resistance to PI3K inhibition [25]. In ovarian cancer cells, the interaction between AXL and EGFR family members or cMET is stimulated by GAS6. It results in the phosphorylation of interacting receptors, ERK pathway activation, and enhanced cell motility [26]. Crosstalk between vascular endothelial growth factor receptor 2 (VEGFR2) and AXL has a different configuration, and involves SRC family kinases (SFK). In endothelial cells, AXL does not phosphorylate VEGFR2. Instead, in vascular endothelial growth factor A (VEGF A)-treated cells, VEGFR2 activates SFK, which phosphorylates AXL at Y779 and Y821 in GAS6-independent manner. This cross-talk stimulates ERK via VEGFR2 and PI3K-AKT signaling through AXL, with both pathways being important for corneal neovascularization [28]. Therefore, the configuration of AXL-regulated molecular networks is cell type-specific and depends on the repertoire of co-expressed receptors, availability of GAS6 or other ligands, and differentiation status of cells.

## 4. AXL Functions in Normal Tissues. Lessons from Mice

Although AXL becomes detectable in embryonic tissues at day 12.5 after fertilization, AXL^−/−^ and TAM triple knockout mice are viable and fertile. However, TAM^−/−^ animals develop various abnormalities after birth, including autoimmune disorders, blindness, and male infertility [29,30].

### 4.1. Defective Efferocytosis in Immune System of TAM^−/−^ Mice Results in Autoimmunity

AXL is present in a wide spectrum of cells in adult tissues, including dendritic cells, macrophages, and platelets [31]. A primary cause of abnormalities in TAM triple knockout animals is the defective clearance of apoptotic cells and their accumulation in mouse tissues. Impaired phagocytosis of apoptotic cells in the immune system is largely responsible for the autoimmune phenotype observed in TAM-deficient mice [32]. The clonal selection of immune cells generates large amounts of dying cells, which undergo clearance by macrophages, dendritic cells, and other phagocytes through efferocytosis. Cells undergoing apoptosis experience modifications of autoantigens, disintegration of their membranes, and leakage of cellular contents (so-called secondary necrosis) [33]. The release of autoantigenic potentially toxic danger signals by dying immune cells, which were not cleared by phagocytes, drives autoimmune reactions. This leads to rheumatoid arthritis and systemic lupus erythematosus in humans, and autoimmune phenotype in TAM-deficient mice [33,34].

### 4.2. Blindness and Male Infertility in TAM Knockout Mice

Similar to the autoimmune phenotype, blindness and male infertility in TAM knockout mice are caused by the defects in efferocytosis. Pathology in the retina of TAM triple knockout mice is caused by functional inactivation of a specialized type of phagocytes containing TAM receptors—retinal pigment epithelial cells (RPE). These cells phagocytose the outer segments of retinal photoreceptors, and thereby remove toxic oxidative phototransduction products. Impairing the function of RPE by TAM knockout leads to widespread apoptosis in the retina, retinal dystrophy, and blindness [34]. A very similar retinal phenotype was observed in mice with combined deletion of genes encoding both TAM ligands, *pros1* and *gas6,* further demonstrating that TAM signaling is indispensable for apoptotic cell clearance in the retina [35]. Pathology in the retina is mostly attributed to the inactivation of MER. Retinal phenotypes are similar in MER^−/−^ and triple TAM^−/−^ mice, but absent in AXL or TYRO3 single knockouts [34].

Phagocytosis in the testis is predominantly carried out by Sertoli cells. These TAM-expressing phagocytes are responsible for the clearance of abundant dead germ cells generated during spermatogenesis. The inactivity of Sertoli cells in TAM^−/−^ mice lead to the accumulation of apoptotic cells in seminiferous tubules where spermatozoa are formed, whereby subsequent male infertility results [32].

### 4.3. AXL Stimulates Efferocytosis by Regulating Cytoskeletal Dynamics

Efferocytosis performed by dendritic cells and macrophages is predominantly dependent on AXL and MER, respectively [36]. GAS6 and PROS1 bind TAM receptors at 2:2 stoichiometry, but the physical availability of the ligand is not sufficient to ensure biological functions of the receptors. N-terminal γ-carboxyglutamic acid-rich (GLA) domain of GAS6 undergoes vitamin K-dependent carboxylation, and this modification stimulates the interaction of the ligand with PtdSer in plasma membranes. In normal circumstances, flippases ensure that PtdSer is located exclusively in the inner leaflets of plasma membranes. However, the cleavage of flippases at early stages of apoptosis results in the externalization of PtdSer in dying cells and apoptotic bodies [37]. Externalized PtdSer represents eat-me signals for macrophages and other phagocytes. AXL and MER exposed on the surface of phagocytes interact with their ligands GAS6 and PROS1 bound to PtdSer-rich membranes of target objects. In addition to TAM receptors and their ligands, other receptors and protein complexes are involved in forming contacts between phagocytes and their targets [38]. Those include direct PtdSer receptors, T-cell immunoglobulin, and mucin domain-containing molecules (TIMs) and αvβ3 or αvβ5 integrins binding to PtdSer-interacting ligand, the milk fat globule-EGF Factor 8 (MFG-E8). The binding of phagocytes to their targets initiates membrane ruffling, formation of longer protrusions with subsequent engulfment of phagocytized objects. Underlying cytoskeletal reorganization is regulated by the concerted action of several small GTPases, including RAC1 and RHOG [39,40]. The guanine exchange factor Dedicator Of CytoKinesis 180 (DOCK180) in a complex with scaffold proteins EnguLfment and MOtility 1 or 2 (ELMO1 or ELMO2) are responsible for the spatiotemporal activation of RAC1, membrane ruffling, engulfment, and phagocytosis [38,40].

### 4.4. AXL Limits the Innate Immune Response

In addition to the stimulation of phagocytosis, TAM receptors, and in particular AXL, protect tissues from autoimmune disorders by limiting the duration of the innate immune response. The activation of Toll-like receptors (TLR) in macrophages and dendritic cells by endogenous danger signals induces an inflammatory response, which is attenuated via the activation of suppressor of cytokine signaling proteins (SOCS) 1 and 3. These inhibitors of Janus kinases/signal transducer and activator of transcription proteins (JAK/STATs) signaling are components of E3 ubiquitin ligase complexes, degrading adaptor molecules, which are required for the TLR pathway activation [41]. In dendritic cells, the expression of SOCS1/3 proteins depends on the presence of AXL, and is driven by a pathway involving STAT1, AXL/IFNα,β receptor (IFNAR) complex and GAS6 [42]. The inactivation of this immunosuppressive pathway, in combination with the defects in efferocytosis, is a key determinant of the autoimmune phenotype in TAM^−/−^ mice.

## 5. AXL and Cancer Cell Motility. Is Efferocytic Machinery Hijacked by Cancer Cells?

### 5.1. Motile Cancer Cells May Utilize DOCK180-ELMO Signaling Implicated in Efferocytosis

In cancerous tissues, apoptotic cells are commonly cleared by the professional phagocytes expressing TAM receptors, namely dendritic cells and macrophages. However, non-professional phagocytes, such as tumour cells, can also engulf and ingest both dying [43,44] and living cells [45]. TAM receptors seem to participate in the non-professional clearance of apoptotic cells in cancer. The ectopic expression of MER in MCF10A and several cancer cell lines stimulate efferocytosis in vitro [44]. AXL is overexpressed in cells of solid tumours suggesting that GAS6-AXL-DOCK180-ELMO pathway may theoretically enable cancer cells to clear tumour tissues from apoptotic corpses. To our knowledge, this assumption has not been addressed so far. However, the data indicate that motile cancer cells may rely on this pathway to drive cytoskeletal dynamics. Indeed, in breast cancer cell lines, AXL physically interacts and phosphorylates ELMO1/2 at two tyrosine residues, and this is required for GAS6-induced RAC1 activation and cell invasiveness [46]. As discussed above, AXL may stimulate signaling pathways via crosstalk with other RTKs, and in addition to GAS6 other ligands, may activate RAC1 via AXL-ELMO-DOCK180 signaling module. At least one example of GAS6-independent and AXL-dependent activation of RAC1 has been reported. The treatment of glioblastoma cells with Hepatocyte Growth Factor (HGF) resulted in cMET-AXL co-clustering, AXL phosphorylation at Y779, recruitment of ELMO-DOCK180 complex, RAC1 activation, reorganization of the cytoskeleton, and enhanced cell motility [47]. The assumption that AXL is involved in different cell motility-inducing pathways, initiated by various growth factors, is in line with in vitro studies demonstrating that targeting AXL by RNA interference or AXL inhibitors diminished cell migration. These observations were made in different cell lines, derived from the pancreatic, breast, bladder, NSCLC, thyroid cancer, and liposarcoma [48].

In addition to AXL-ELMO-DOCK180, other AXL-activated pathways stimulate cell motility and invasiveness in vitro. In particular, in hepatocellular carcinoma cells, AXL induces cell migration via activation of PI3K, and subsequent AKT- and GTPase-independent stimulation of p21 (RAC1) activated kinase 1 (PAK1), a critical kinase implicated in cytoskeleton remodeling and control of directional motility [49]. In addition to the direct regulation of cytoskeletal processes, AXL was reported to stimulate cell motility by increasing the turnover of focal adhesions [50]. Mechanistically, this novel pathway represents direct phosphorylation of the scaffold protein, neural precursor cell expressed, developmentally down-regulated 9 (NEDD9) by AXL, the recruitment and phosphorylation of other signaling proteins, such as Paxillin, and destabilization of cell adhesions leading to motile cell phenotype. Understanding whether the elements of this pathway are also implicated in efferocytosis downstream of integrins and TAM receptors remains to be explored.

### 5.2. Are Exosomes in Tumour Microenvironment Involved in AXL-Induced Cell Migration?

The activation of AXL signaling by PtdSer-containing membranes is not limited to apoptotic cells. Apoptotic debris is not the sole source of externalized PtdSer in both normal and pathological conditions. Small extracellular vesicles (sEVs) or exosomes are roughly 100 nm particles produced by platelets, other types of normal cells, as well tumour cells. sEVs play an important role in intercellular communications in normal tissues, and in tumour-stroma crosstalk in cancer [51]. Outer leaflets of sEVs membranes are enriched for PtdSer [52], which is compatible with the hypothesis that TAM receptors are involved in signaling pathways initiated by sEVs [53]. Indeed, sEVs isolated from serum promoted migration of prostate, colon, lung, or breast cancer cells in vitro by activating TYRO3 in the presence of PROS1 [54]. sEVs-activated migration via TYRO3 is mediated by RHOA-ROCK2 pathway, and inactivation of cofilin via phosphorylation. Studies have not yet been conducted to explore whether the availability of ligands determines which TAM receptor is activated by sEVs. Perhaps, in the bloodstream where the concentration of PROS1 is much higher than GAS6 [55], TYRO3 is primarily responsible for sEVs-induced pathways in circulating tumour cells. It is possible that in tumour microenvironment, where the level of GAS6 is high [56,57], AXL takes a central stage.

## 6. AXL and Drug Resistance in Cancer

### 6.1. Survival in Toxic Conditions (Analogy with Professional Phagocytes)

The clearance of apoptotic cells often occurs in toxic environments where professional phagocytes must survive and perform their functions. The engagement of TAM ligands with PtdSer on the surface of apoptotic cells allows phagocytes to sense toxicity and activate pro-survival mechanisms, including canonical PI3K-AKT and RAS-MAPK pathways. By hijacking TAM-activated pathways, cancer cells acquire the capability to survive when they detect early signs of apoptosis in the environment. Accordingly, numerous studies identify TAM-initiated pathways, and in particular, GAS6/AXL signaling as a mechanism of acquired resistance to chemo-, radio-, immune-, and targeted therapies. As AXL has been recognized as a prospective therapeutic target, several small molecule inhibitors (AXLi) with various degrees of selectivity or antibody-based inhibitors have been developed [3,4,58]. AXL inactivation occurs, either by the inhibitors or through RNA interference sensitized cancer cells to gamma-irradiation, antimitotic, and DNA damaging compounds in vitro. The application of AXLi-enhanced cytotoxic and anti-tumour effects of compounds in vitro target RTKs, EGFR, HER2, PDGFR, cMET, VEGFR, and cKIT [48,59]. Moreover, AXL inhibition has a strong potential to overcome therapy resistance in vivo. This has been shown in various preclinical models of solid tumours, including breast, lung, pancreatic, ovarian, esophageal, head and neck cancer, malignant melanoma, and brain tumours. Targeting AXL were shown to sensitize tumours to various types of therapies, including DNA damaging agents, EGFR, VEGFR, HDAC, G2/M checkpoint, and immune checkpoint inhibitors (Table 1).

### 6.2. Protection of Healthy Tissues from Autoimmune Damage and Immune Evasion in Cancer: Common Mechanisms

Immune checkpoint blockade by PD-1, PD-L1 or CTLA-4 inhibitors improves survival in a proportion of patients with solid tumours. However, most of the patients do not respond to the immunotherapy or develop resistance caused by the immunosuppressive tumour microenvironment (TME). The activity of innate immune cells is required to stimulate the cytotoxic function of T lymphocytes, and insufficiency of dendritic cells may cause the failure of immune surveillance. TAM receptors, whose normal function is to limit the duration of innate immune response, mediate immune suppression in TME. Specifically, as discussed in a previous section, AXL pathway terminates TLR signaling in dendritic cells via SOCS1/3. This reduces the secretion of pro-inflammatory cytokines and counteracts tumour infiltration by T cells [36,37].

In addition to AXL function in professional phagocytes, AXL expression in tumour cells is yet another factor contributing to immunosuppression. In breast cancer cells, AXL pathway activation results in decreased expression and presentation of MHC class I antigens, leading to the decreased tumour infiltration by CD4^+^ and CD8^+^ T cells, and consequently, immune evasion [75,76]. In line with these findings, another important study has shown that anti-PD1 therapy-resistant melanomas expressed high levels of AXL in cancer cells [77]. Similarly, AXL signaling correlates with the elevated expression of PD-L1 in EGFR-mutant NSCLC cells [78]. In addition to driving negative feedback regulation in dendritic cells, AXL and other TAM receptors control functions of NK cells, another critical component of the innate immune system. In particular, NK cells attack and eliminate those cancer cells, which are deficient in the expression of MHC class I antigens. TAM-driven pathways downregulate the expression and function of the receptors, which operate during the differentiation and maturation of NK cells [79]. By suppressing the activity of NK cells, TAM signaling promotes metastatic spread in mouse models of malignant melanoma and breast cancer [80].

Thus, TAM receptors inhibit innate immunity leading to the reduced infiltration of CD4^+^ and CD8^+^ lymphocytes and incapacitate NK cells. Therefore, disabling TAM function sustains immunostimulatory TME, which may improve the efficacy of immune checkpoint inhibition [4]. Indeed, selective small molecule AXLi synergizes with an PD-1-blocking antibody to inhibit the growth of ovarian cancer and glioblastoma in xenograft models [9,62]. Moreover, another approach for combined targeting AXL and immune checkpoint has shown promise. An AXL-specific antibody-drug conjugate Enapotamabvedotin (EnaV) was generated by crosslinking anti-AXL IgG1 with auristatin E, a cytotoxic agent acting via destabilization of microtubules [71]. In xenograft melanoma and lung cancer mouse models, combined application of EnaV and anti-PD-1 therapy significantly prolonged the survival of mice as compared with single treatments [73] (Table 1).

## 7. AXL, Epithelial-Mesenchymal Plasticity, Drug Tolerant Persister Cells, and Cancer Metastasis

### 7.1. Cancer Cells Exist in Distinct Differentiation States

As discussed in the above sections, AXL-expressing tumour cells are endowed with invasive and drug resistant characteristics. These features are hallmarks of epithelial-mesenchymal transition (EMT), an embryonic genetic program hijacked by cancer cells. EMT and the reverse process, mesenchymal-epithelial transition (MET), constitute a major source of tumour cell plasticity, and represent an important factor of cancer heterogeneity [81,82]. Notably, the transcription factors that regulate EMT (so-called EMT-TFs) during normal embryonic development and tumorigenesis are shared. EMT-TFs belonging to the ZEB (ZEB1 and ZEB2), SNAIL (SNAIL1 and SNAIL2/SLUG), and TWIST (TWIST1 and TWIST2) families are best-studied in the context of cancer [83,84]. EMT-TFs repress transcription of epithelial markers, and directly or indirectly activate mesenchymal genes such as vimentin. In the course of complete mesenchymal reprogramming, cells lose all epithelial characteristics, gain mesenchymal markers, mesenchymal type of cell polarity, and high invasive capabilities. It is now broadly accepted that EMT and MET are not binary processes, and cancer cells acquire intermediate differentiation states combining epithelial and mesenchymal features (so-called partial or hybrid EMT) [85,86] (Figure 2A). These cell populations with hybrid epithelial and mesenchymal characteristics were isolated from genetically manipulated mouse models and patient-derived xenotransplants [85,87]. Whereas proliferative potential gradually decreases when cells progress from epithelial to a mesenchymal state, their invasive potential increases. Cells in a hybrid EMT state express EMT-TFs; they exhibit the highest tumourigenic potential and metastatic capacity. Hybrid (especially late hybrid) EMT cells are phenotypically less stable than fully differentiated epithelial or mesenchymal tumour cells. This can be explained by the fact that epithelial or mesenchymal endpoints are stabilized by self-enforcing double-negative feedbacks formed between EMT-TFs and certain miRNA species (SNAIL1-miR-34 or ZEB1/2-miR-200) [85].

### 7.2. AXL Belongs to the Mesenchymal Gene Expression Signatures Characterizing Aggressive Cancers

The association between AXL signaling and EMT has been reported in different cancer types. The expression and function of AXL are regulated at the transcriptional level by proteins implicated in epithelial-mesenchymal plasticity. ΔNp63α, HIF1α, YAP1/TEAD and FRA1/cJUN transcription factor complexes, and EMT-TF ZEB1 control *AXL* gene transcription directly [88,89,90,91,92]. At the post-transcriptional level, AXL expression is co-regulated with EMT-TFs of SNAIL family by miR-34, a p53-induced SNAIL repressor [93,94,95]. Transcription factors up-regulating *AXL* transcription are parts of gene regulatory networks operating in mesenchymal and hybrid states. Whereas, miR-34 favours MET in cells with wild-type p53. Thus, *AXL* seems to belong to gene expression patterns favouring EMT and invasion, whose activation is promoted by mutations in *TP53*. In line with these considerations, AXL expression was detected in the most aggressive subtypes of bladder and ovarian cancers characterized by EMT and mutant *TP53* signatures [2,26,96,97,98]. In breast cancer, AXL was identified as a component of the ΔNp63α-driven hybrid EMT program. The resulting cells retained some epithelial traits, but were highly invasive [90]. In several reports, a driver rather than a passenger role of AXL in EMT has been demonstrated. AXL inhibition by small molecule compounds or RNA interference led to a partial MET in breast, lung, ovarian, or pancreatic carcinoma cells [1,26,99,100].

Interestingly, while AXL is expressed in mouse embryos, it is implicated in cancer EMT and EMT drives critical stages of embryonic development. AXL expression is dispensable for embryogenesis. This peculiarity distinguishes AXL from many other RTKs, as well as most EMT inducers and effectors whose genetic ablations have more severe phenotypes and often cause embryonic lethality. The lack of embryonic abnormalities in AXL^−/−^ mice resembles normal development of mouse embryos with depleted vimentin, a canonical EMT marker [101].

### 7.3. AXL Is Implicated in Metastasis in Experimental In Vivo Models

Consistent with the role of AXL in promoting EMT in cancer, targeting AXL by selective inhibitors, germline knockout, RNAi, and inactivating mAbs prevented etastasis of various epithelial tumours, including but not limited to, breast, lung, ovarian, and colorectal cancers. The analysis has been carried out in genetic mouse models, cell line-derived xenografts (CDX), patients-derived xenografts (PDX), and in various experimental metastasis assays (summarized in Table 2).

Although AXL inhibition prevents metastases (Table 2), constitutive ectopic expression of AXL in metastatic breast cancer cells did not promote, but instead reduced the metastatic traits analysed in intravenous metastasis assay. In later stages of the metastatic process, the downregulation of AXL signaling was required for the interaction between cancer cells and the microenvironment in the metastatic niches, and successful metastatic colonization of the lungs [104]. Therefore, the reversibility of AXL signaling, but not persistent AXL activation facilitates metastatic dissemination. Two considerations support this conclusion. Firstly, the *AXL* gene is rarely amplified or mutated in human cancer, which is in stark contrast to some other RTKs. Secondly, the expression of AXL is a characteristic of cells residing in the hybrid state of EMT spectrum [90,120]. This cell state retains the potential to regenerate fully differentiated AXL-negative cells.

### 7.4. Hypothetical Role of AXL in the Formation of Drug Tolerant Persister Cancer Cells

A dynamic equilibrium between differentiation states along the EMT/MET axis exists in most solid tumours (Figure 2A). EMT-TFs are present in cells with hybrid or mesenchymal phenotype, and activate mechanisms maintaining genome integrity, multi-drug resistance, and cell survival [121]. The survival of tumour cells with mesenchymal or hybrid characteristics upon treatment with conventional chemotherapy drugs has been demonstrated in vivo using mouse models of breast and pancreatic adenocarcinoma [122,123]. Acute exposure of cancer cells to the drugs in vitro does not lead to the eradication of the whole cell population. A small pool of viable cells, or DTPs, survive treatment and represent the source for the genetic evolution of tumours and eventually relapse [124] (Figure 2B). The presence of DTPs in the bone marrow of cancer patients after therapeutic intervention, a condition termed the minimal residual disease, indicates a high probability of cancer relapse. It has been previously hypothesized that DTP cells originate from the tumour cells residing in hybrid or mesenchymal states along the EMT/MET spectrum [125,126].

It can be speculated that AXL pathways play a decisive role in the generation and survival of DTP pools. Application of cytotoxic drugs induces apoptosis in the epithelial cells representing the bulk of the tumour, and externalization of PtdSer in the affected cells. The engagement of PtdSer with GAS6 may lead to the activation of AXL signaling in hybrid EMT/MET tumour cells (Figure 2). Within TME, GAS6 is broadly available; it is secreted by some tumour cells, cancer-associated fibroblasts (CAFs), dendritic cells, and tumour associated macrophages. Production of GAS6 is further stimulated by immunosuppressive cytokines IL10 and IFNα, whose presence in TME is promoted by externalized PtdSer in apoptotic cells and bodies [127]. Activation of AXL signaling in hybrid EMT/MET tumour cells will have several important consequences. First, the signaling contributes to cell survival via canonical anti-apoptotic pathways leading to the formation of drug-resistant DTP pools. Secondly, the activation of AXL in hybrid cells moves the EMT/MET equilibrium towards the mesenchymal end of the EMT/MET spectrum. Activation of the AXL-ELMO-DOCK180 axis and other EMT-associated signaling pathways lead to the enhanced migratory and invasive potential of DTP cells, their ability to disseminate and switch on the immune-escape mechanisms [128]. This hypothetical model explains broadly documented implication of AXL pathway in drug resistance and cancer metastasis (Table 2).

## 8. Concluding Remarks

AXL signaling has emerged as an important pathway contributing to tumour progression, metastasis, and therapy resistance. This critical role of AXL in various types of solid cancers arises from AXL function in healthy tissue. Mouse modelling and in vitro experiments demonstrated that AXL, and its relative MER, are essential for the activity of professional phagocytes in different tissues. The engulfment of dead cells during efferocytosis involves the activation of PI3K signaling, cytoskeletal reorganization via ELMO-DOCK180, and other molecular pathways implicated in cancer cell motility. The efficient clearance of apoptotic cells is an important factor of anti-inflammatory response. In addition, the role of AXL in immunosuppression is associated with the down-regulation of TLR signaling in dendritic cells and suppression of NK activity.

The AXL receptor is expressed both in tumour cells, which experience a complete or partial EMT, and in TME. It seems that in tumour cells, AXL signaling is activated via a mechanism similar to that operating during efferocytosis, when GAS6 ligand binds PtdSer on the membranes of dying cells and apoptotic bodies. Both these factors, GAS6 and PtdSer, are readily available in TME. GAS6 is produced by CAFs and other cells within tumour tissue, and externalized PtdSer is abundant due to stress- and cell death-causing conditions, such as oxygen radicals or hypoxia. Therapeutic interventions induce cell death in bulks of the tumours, and thereby promote AXL activation in subpopulations of AXL-expressing cells. This leads to the activation of cell survival via PI3K and EMT pathways and the formation of pools of DTP cells, which are therapy-resistant and invasive. 

Another feature that specifies a broad role of AXL pathway in cancer biology is the incorporation of AXL in oncogenic signaling pathways operating in various cancers. AXL physically and functionally interacts with EGFR family members, other RTKs, and SRC family kinases among others. The consequence of this deep integration of AXL in oncogenic signaling networks is GAS6-independent activation of AXL in some cases.

Equally important for tumour biology is AXL function in dendritic and NK cells. By terminating TLR signaling in dendritic cells and suppressing NK, AXL generates an immunosuppressive environment and helps cancer cells to evade immune surveillance. The dual role of AXL in tumour and immune cells explains the reason why AXL inhibition synergizes with such a broad spectrum of therapeutic agents. The targeting of AXL moves cancer cells toward the epithelial end of EMT/MET spectrum and sensitizes them to chemo-, radio- or targeted therapies, which efficiently eradicate proliferating epithelial cells. On the other hand, AXL inactivation blocks the negative regulation of dendritic and NK cells, stimulates innate immunity, and thereby potentiates the efficacy of immune checkpoint inhibitors (Table 1). Currently, ongoing clinical trials have been aimed at identifying combinatorial therapeutic schemes which include AXL inhibition and are optimal for particular cancer types. Further research is needed to discover biomarkers that will help identify benefits to patients and evaluate tumour responses to AXL-targeted therapies.

## Figures and Tables

**Figure 1 cancers-13-04864-f001:**
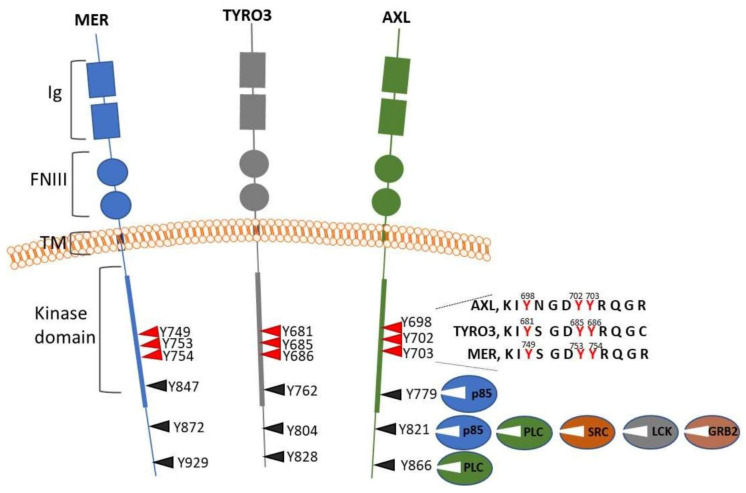
A scheme portraying TYRO3, MER and AXL receptors. Conserved domains, fibronectin type III (FNIII), immunoglobulin (Ig)-like, transmembrane (TM), and kinase domains are shown. Phosphorylated tyrosines within kinase domains and distal cytoplasmic regions are indicated. The scheme shows SH2-containing docking proteins interacting with AXL C-terminal phosphotyrosines. p85, regulatory PI3K subunit; PLC, phospholipase C; SRC, cellular tyrosine kinase c-SRC; LCK, lymphocyte-specific protein tyrosine kinase; GRB2, growth factor receptor-bound protein 2.

**Figure 2 cancers-13-04864-f002:**
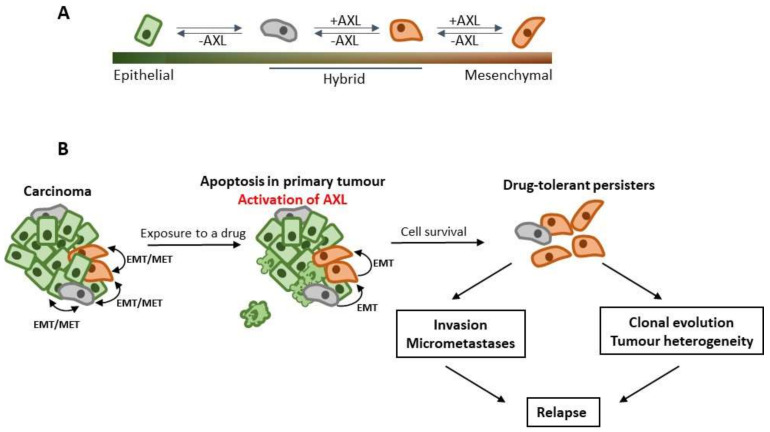
AXL pathway promotes drug resistance and metastasis in carcinoma. (**A**) Reversible EMT states of tumour cells in carcinomas. Transient activation of AXL signaling shifts the balance towards the mesenchymal end of the spectrum. (**B**) Therapeutic intervention results in apoptosis in a bulk of a tumour leading to PtdSer externalization and activation of AXL pathway in a proportion of tumour cells. By activating cell survival and EMT pathways, AXL stimulates the formation of invasive and drug resistant cells (drug tolerant persisters, DTPs). DTPs represent the source for the genetic evolution of cancer, cancer relapse, and spread.

**Table 1 cancers-13-04864-t001:** Targeting AXL signaling sensitizes solid tumours to various therapies in vivo.

Cancer Type	Type of Study	Method of AXL Inhibition	Therapeutic Agent	Drug Class	Reference
Breast cancer	(*) CDX	mAb (YW327.6S2)	Anti-VEGF mAb	VEGFR-Targeted therapy	[60]
Breast cancer	Cell lines-derived xenografts (CDX)	AXL shRNA, AXLi (R428)	Doxorubicin	Chemotherapy	[61]
Ovarian cancer	Mouse CDX	AXLi (R428)	Anti-PD-1 mAb	Immunotherapy	[62]
Lung cancer	CDX	mAb (YW327.6S2)	Erlotinib	EGFR-Targeted therapy	[60]
Lung cancer	CDX	mAb (YW327.6S2)	Paclitaxel + Carboplatin	Chemotherapy	[60]
Lung cancer, NSCLC	CDX	AXL shRNA, AXLi	Erlotinib	EGFR-targeted therapy	[63]
Lung cancer, SCLC	CDX	AXLi (TP0903)	AZD1775	WEE1-targeted therapy	[64]
Lung cancer, breast cancer	Mouse CDX, orthotopic model for breast cancer	TAMi (Sitravatinib)	Anti-PD-1 mAb (Nivolumab)	Immunotherapy	[65]
Lung cancer, NSCLC	CDX, (**) PDX	AXLi (NPS1034)	Osimertinib,	EGFR-targeted therapy	[66]
Lung cancer, NSCLC	Human CDX	AXLi (ONO-7475)	Osimertinib	EGFR-targeted therapy	[67]
Esophageal adenocarcinoma	Mouse CDX	AXLi (R428)	Epirubicin	Chemotherapy	[68]
Pancreatic cancer	Transgenic model (Kras^LSL-G12D^; Cdkn2a^lox/lox^; Ptf1a^Cre/+^); orthotopic model	AXLi (R428)	Gemcitabine	Chemotherapy	[69]
Head and Neck Cancer	PDX	AXLi (R428)	Cetuximab or radiation	EGFR-targeted therapy or radiotherapy	[70]
Cutaneous melanoma	PDX	(***) AXL-107-MMAE (or EnaV)	Vemurafenib + Trametinib	BRAF^V600E^ + MEK − targeted therapy	[71]
Cutaneous melanoma	PDX	AXLi (R428)	AZD7762	CHK1/CHK2-targeted therapy	[72]
Cutaneous melanoma; lung cancer	CDX, PDX	(***) AXL-107-MMAE (or EnaV	Anti-PD-1 mAb (Pembrolizumab)	Immunotherapy	[73]
Glioblastoma	Mouse spheroid-derived xenografts	AXLi (R428)	Anti-PD-1 mAb (Nivolumab)	Immunotherapy	[9]
Diffuse intrinsic pontine glioma	PDX; mouse allografts	AXLi (R428)	Panobinostat	Histone deacetylase inhibitors	[74]

(*) CDX, cell lines-derived xenografts. (**) PDX, patient-derived xenografts. (***) AXL-107-MMAE, an anti-AXL antibody-drug conjugate. It represents a fusion between mAb and the microtubule-disrupting agent monomethyl auristatin E.

**Table 2 cancers-13-04864-t002:** In vivo models demonstrate essential role of AXL in cancer metastasis.

Cancer Type	Type of Study	Method of AXL Inhibition	Reference
Breast cancer	Orthotopic model	shRNA	[102]
Breast cancer	Intravenous metastasis assay	Anti-AXL mAb YW327.6S2	[60]
Breast cancer	Orthotopic model and lung metastasis assay	AXLi R428	[103]
Breast cancer	Lung metastasis assay	AXLi R428; shRNA	[104]
Breast cancer	(*) CDX	AXLi R428; shRNA	[101]
Breast cancer	CDX	AXLi	[105]
Breast cancer	CDX	AXLi	[106]
Breast cancer	Orthotopic model	CRISPR/Cas9 gene inactivation	[107]
Triple-Negative Breast Cancer	Orthotopic model	shRNAi	[99]
Triple-Negative Breast Cancer	CDX or (**) PDX	Anti-AXL mAb 20G7-D9	[108]
Triple-Negative Breast Cancer	CDX	pan-TAM kinase inhibitor BMS-777607	[109]
HER2 + Breast Cancer	Genetically modified mice	Germline knockout	[27]
Breast cancer, cutaneous melanoma	CDX	Pan-TAM inhibitor LDC1267	[80]
Ovarian, breast, pancreatic cancer	CDX	MYD1-72 Fc decoy receptor	[110]
Ovarian cancer	Peritoneal xenografts	shRNA, sAXL (acts as a decoy receptor)	[111]
Ovarian cancer	Intraperitoneal injections	MYD1 Fc decoy receptor	[112]
Ovarian cancer	CDX	AXL-aptamer	[113]
Endometrial cancer	Orthotopic	shRNA	[114]
Uterine cancer	CDX	siRNA	[115]
Pancreatic cancer	CDX	Anti-AXL mAb 10C9	[100]
Colorectal cancer	CDX (effect on dissemination in the bloodstream)	Ectopic expression of AXL	[116]
Gastric cancer	CDX	shRNA	[117]
Clear cell renal cell carcinoma	Lung metastasis assay	shRNA, sAXL-IgG1 fusion	[91]
NSCLC	Lung metastasis assay	shRNA	[118]
NSCLC	Intracardiac injections	shRNA	[119]

(*) CDX, cell lines-derived xenografts. (**) PDX, patient-derived xenografts.

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
