# Peer review of "AXL Receptor in Cancer Metastasis and Drug Resistance: When Normal Functions Go Askew"

_cancers, 2021, doi:10.3390/cancers13194864_

Round 1

Reviewer 1 Report

The manuscript entitled "AXL receptor in cancer: when normal functions go askew." tried to cover as much of the recent data related to Axl as possible and to summarize and find to the relevance in many aspects of cell biology and oncology.

I recommend to resubmit the manuscript for publication Cancers with revision of the following concerns.

Major concerns;

  1. In Abstracts, the contents reviewed in this article were not well summarized. For example, only cancer-related content is mentioned. There is on
  2. In section 2 (overview of the structure of TMA receptor), TAM receptors and various ligands are described out of sequence. It would be better to separate each of them completely and change the title of section 2.
  3. In section 4 (Normal Axl function) and section 5 (efferocytosis…..), it would be better to merge the contents of each section and reorganize sections according to the Axl-related phenotypes such as efferocytosis, cytokinesis, and cell migration etc.
  4. In section 6 (Axl and drug resistance), the first two sentences (line 268 ~ 272) do not seem to be needed.
  5. It would be better to merge Section 6, 7 and 8. And then the contents should be separate into Axl-targeted cancer therapy, intervention of chemo-resistance and its involvement of EMT and metastasis.
  6. It seems to be better to reorganize the conclusion section to cover the contents reviewed in previous sections.

Minor concerns;

  1. There is no full name of SRC. (page 3, line 113)
  2. There are some typos. (For example, GAS)

Author Response

We thank both reviewers for their useful and constructive criticism. We amended the manuscript according to the reviewers’ comments.

In addition, we introduced the subsections into the manuscript, which we believe makes our review more reader-friendly.   

We used Grammarly program to correct grammar and style.

REVEWER 1

I recommend to resubmit the manuscript for publication Cancers with revision of the following concerns.

Major concerns;

In Abstracts, the contents reviewed in this article were not well summarized. For example, only cancer-related content is mentioned. There is on

  • The Abstract is modified according to this comment. In the new version, the Abstract contains the information on normal AXL functions.

In section 2 (overview of the structure of TMA receptor), TAM receptors and various ligands are described out of sequence. It would be better to separate each of them completely and change the title of section 2.

  • We structured this section and subdivided it into two subsections. The title of this section is changed.

In section 4 (Normal Axl function) and section 5 (efferocytosis…..), it would be better to merge the contents of each section and reorganize sections according to the Axl-related phenotypes such as efferocytosis, cytokinesis, and cell migration etc.

  • According to this comment, we discuss normal AXL functions in one section of the revised version of the paper (section 4). This section summarizes AXL functions in efferocytosis in different normal tissues, and data on the regulation of innate immunity.

In section 6 (Axl and drug resistance), the first two sentences (line 268 ~ 272) do not seem to be needed.

  • In this paper, we attempt to show how cancer cells exploit normal AXL-regulated cellular processes to develop invasive phenotype and resistance against apoptotic insults. In these sentences, we propose that normal cells operating in toxic environment (such as professional phagocytes) activate cell survival pathways, which are also utilised by AXL-expressing cancer cells. Therefore, we prefer to keep these sentences in the text (subsection 6.1 in the revised version).

It would be better to merge Section 6, 7 and 8. And then the contents should be separate into Axl-targeted cancer therapy, intervention of chemo-resistance and its involvement of EMT and metastasis.

  • We discuss the mechanisms of drug resistance in cancer and their relationship with normal AXL functions in section 6. We merge sections 7 and 8 to discuss AXL implication in epithelial-mesenchymal plasticity and generation of DTP pools.

It seems to be better to reorganize the conclusion section to cover the contents reviewed in previous sections.

  • According to this comment, we reorganised conclusion section. In the new version, this section contains information on AXL functions in normal tissues.

Minor concerns;

There is no full name of SRC. (page 3, line 113)

There are some typos. (For example, GAS)

  • The abbreviation “SRC” is explained in the text and in Figure 1 legend of the revised paper. The indicated typo is corrected. Several additional errors were revealed and corrected.

Reviewer 2 Report

Comment 1. The title should include “metastatic and drug-resistant” to specify their review.

Comment 2. Lines 158: danger signals should be defined.

Comment 3. Table 1 and 2 are hardly legible. Please define CDX and PDX in Table legends (not in Table). 

Comment 4. Line 412: “In in vitro experiments”, please remove double “in”.

Comment 5. Figure 2B: It would be more informative for readers to understand the relationship among clonal evolution, relapse and heterogeneity. I recommend creating a schematic diagram more accurately.

Author Response

We thank both reviewers for their useful and constructive criticism.

We amended the manuscript according to the reviewers’ comments. In addition, we introduced the subsections into the manuscript, which we believe makes our review more reader-friendly.   

We used Grammarly programme to correct grammar and style.

REVIEWER 2

Comments and Suggestions for Authors

Comment 1. The title should include “metastatic and drug-resistant” to specify their review.

  • The title of the revised manuscript contains words “cancer metastasis” and “drug resistance”.

Comment 2. Lines 158: danger signals should be defined.

  • We define danger signals as “autoantigenic potentially toxic … signals” released by dying immune cells and provide a reference to a review paper on this subject [33].

Comment 3. Table 1 and 2 are hardly legible. Please define CDX and PDX in Table legends (not in Table).

  • In the revised version, these abbreviations are defined in the legends.

Comment 4. Line 412: “In in vitro experiments”, please remove double “in”.

  • This typo is corrected

Comment 5. Figure 2B: It would be more informative for readers to understand the relationship among clonal evolution, relapse and heterogeneity. I recommend creating a schematic diagram more accurately.

  • We are particularly grateful to this reviewer for bringing this inaccuracy to our attention. Figure 2B is corrected in the revised version of the manuscript.

Round 2

Reviewer 1 Report

The author has adequately answered both major and minor comments point by point. I recommend to accept the manuscript for publication Cancers.  

Reviewer 2 Report

Accepted.